

# What can seismic noise tell us about the Alpine reactivation of the Iberian Massif? An example in the Iberian Central System

Juvenal Andrés[1,2], Puy Ayarza[2], Martin Schimmel[1], Imma Palomeras[2], Mario Ruiz[1], and Ramon Carbonell[1]

[1]Institute of Earth Science Jaume Almera (ICTJA), 08028, Barcelona, Spain
[2]Department of Geology, University of Salamanca, 37008, Salamanca, Spain

*Correspondence to*: Juvenal Andrés (juvenalandrescabrera@gmail.com) /Ramon Carbonell (ramon.carbonell@csic.es)

**Abstract.** The Iberian Central System, formed after the Alpine reactivation of the Variscan Iberian Massif, features maximum altitudes of 2500 m. It is surrounded by two foreland basins with contrasting elevation: The Duero Basin to the N, located at 750-800 m and the Tajo Basin to the S, lying at 450-500 m. The deep crustal structure of this mountain range seems to be characterized by the existence of a moderate crustal root that provides isostatic support for its topography. New seismic data is able to constrain the geometry of this crustal root, which appears to be defined by a northward lower crustal imbrication of the southern Central Iberian crust underneath this range. Contrarily to what was expected, this imbrication also affects the upper crust, as the existing orogen-scale mid-crustal Variscan detachment was probably assimilated during the Carboniferous crustal melting that gave rise to the Central System batholith. This implies that the reactivated upper crustal fractures can reach lower crustal depths, thus allowing the entire crust to sink. This new model can explain the differences in topography between the Central System foreland basins. Also, it provides further constrains on the crustal geometry of this mountain range, as it seems to be that of an asymmetric Alpine-type orogen, thus hindering the existence of buckling processes as the sole origin of the deformation. Results presented here have been achieved after autocorrelation of seismic noise along the CIMDEF profile. Although the resolution of the dataset features limited resolution (0.5-4 Hz, stations placed at ~5 km), this methodology has allowed us to pinpoint some key structures that helped to constraint the deformation mechanisms that affected Central Iberia during the Alpine orogeny.

## 1 Introduction

The Iberian Central System (ICS) is an intraplate mountain range that divides the Iberian Central Meseta in two sectors – the northern Duero Basin (DB) and the Tajo Basin (TB) to the S (Fig. 1, Andrés et al., 2019). The most striking feature of the Central Meseta is its highly variable topography, with the Tajo Basin having an average altitude of 450-500 m while the Duero Basin presents a higher altitude of ~750-800 m. It is thought that this contrast in altitude of about 300 m should mainly respond to subsurface characteristics (e.g. crustal structure or rheological properties of the lithosphere) but its origin is, as yet, unknown. The ICS range acts as a boundary between the two basins, that are enclosed within the Iberian Massif (IB) and extends in a NE-SW to ENE-WSW direction across the Iberian Peninsula for over 300 km, with some peaks reaching 2500 m.





The lithospheric structure of the IB has been largely studied since the 1990's by different seismic techniques, e.g., controlled source seismic studies (Banda et al., 1981; ILIHA DSS Group, 1993, Pulgar et al., 1996, Ayarza et al., 1998, 2004; Suriñach and Vegas, 1998; Simancas et al., 2003; Carbonell et al., 2004; Flecha et al., 2009; Palomeras et al., 2009, 2011; Martínez

Poyatos et al., 2012; Ehsan et al., 2014; 2015), receiver functions (Mancilla and Diaz, 2015), shear wave tomography (Palomeras et al., 2017), etc. Also, gravity modelling (e.g., de Vicente et al., 2007, Torne, et al., 2015) has been used to unravel the crustal structure of the lithosphere of the Iberian microplate. Most of these studies have focused mainly on the southern and northern parts of the IB. Only the regional study of de Vicente et al. (2007) and the large scale studies of Mancilla and Diaz (2015), Diaz et al. (2016), Palomeras et al. (2017) and Torne et al. (2015) cover the ICS, the DB and the TB. However,

the lack of detailed seismic data on the centre of the IB hinders the complete comprehension of the current lithospheric structure of the ICS, its evolution and that of its bounding basins.

To overcome this problem, the CIMDEF project was designed to acquire natural and controlled source seismic data across the ICS, the DB and the TB. The resulting data can be integrated with the existing models and provide a complete section of the

IB. As the first result of this effort, Andrés et al. (2019) presented a lithospheric model of the area using Global Phase Seismic Interferometry (GloPSI) of teleseismic data (Ruigrok and Wapennar, 2012). In this paper, ambient seismic noise recordings are used to construct a lithospheric profile coincident with that of Andres et al. (2019). Ambient noise has lately proven to be a useful and inexpensive tool for lithospheric imaging. Recent studies have exploited the recorded ambient noise field (e.g., Tibuleac & von Seggern, 2012; Gorbatov et al., 2013; Taylor et al., 2016; Kennett et al., 2015, 2016; Becker & Knapmeyer-

Endrun, 2018; Buffoni et al., 2019) to image lithospheric discontinuities such as the Moho, the Hales discontinuity or the Lithosphere-Astenorsphere Boundary (LAB). These studies rely on the construction of autocorrelograms of the recorded ambient seismic field as they retrieve the Green's function of the response of the subsurface structure.

This work aims to contribute to the knowledge on the crustal structure and crustal thickness across the ICS and its relationship

with the DB and TB. The results will provide us with constraints that are an asset to study the origin and evolution of the topography and the intraplate deformation dynamics of Central Iberia during the Alpine Orogeny. The observations presented in this work are derived from the CIMDEF experiment and present a continuation and additional support to those of Andrés et al. (2019), complementing and extending the previous knowledge of the ICS orogen.

## 2 Geological Setting

The basement of the Iberian Peninsula is composed by Upper Proterozoic to Carboniferous rocks deformed and intruded by granites mainly as a consequence of the Variscan Orogeny. The orogeny took place during the Late Paleozoic times by the



collision between Laurussia and Gondawa (Matte, 2001), which closed the Rheic Ocean and amalgamated these continents along with other minor terranes like Armorica (Franke, 2000; Matte, 2001).

The IB, the Iberian outcrop of the European Variscides, is subdivided in six zones (Fig.1) (Lotze, 1945; Julivert et al., 1972; Farias et al., 1987; Arenas et al., 1988), from N to S being the Cantabrian Zone (CZ), the West Asturian-Leonese Zone (WALZ), the Galicia Tras-Os-Montes Zone (GTMZ), the Central Iberian Zone (CIZ), the Ossa-Morena Zone (OMZ), and the South Portuguese Zone (SPZ). The CZ and the SPZ are interpreted as the foreland fold and thrust belts, while the rest of the areas correspond to the internal zones of the orogen. The GTMZ is an allochthonous unit which overlies the CIZ, and it is

composed by Gondwana terrains and ophiolites corresponding to vestiges of oceanic crust with high-pressure metamorphism (Martínez Catalan et al., 2014). These rocks, together with those found between the SPZ, OMZ, and CIZ, suggest the existence of one or more sutures (Simancas et al., 2013). Three out of these six domains, the CIZ, CZ and WALZ, represent continental portions of the passive margin along Gondwana before the Variscan orogeny. The significance of the CIZ, the widest of all zones, is currently under discussion. Some authors have therein defined the Central Iberian Arc (CIA) (Martínez Catalán,

2011a, 2011b), which, together with the Ibero-Armorican Arc, would define a double Variscan orocline. However, Pastor-Galán et al. (2015, 2016, and 2017) argue that it is not a double orocline but a curve originated as a combination of processes occurred in Variscan times and later in the Cenozoic, during the Alpine tectonics. Overlying the CIZ, the GTMZ is a relic of the Rheic Ocean formed partly by ophiolites. The OMZ is interpreted as a ribbon continental domain that drifted to some extent from Gondwana. Finally, the SPZ is interpreted as a fragment of Laurussiaº.


The CIMDEF passive seismic profile is located within the CIZ (Fig. 1). The latter is subdivided in two zones (Díez Balda et al., 1990): The Ollo de Sapo Domain to the N, and the Schist-Greywacke Complex to the S. The first is characterized by high-grade metamorphism, high deformation (Barbero and Villaseca, 2000) and a great volume of outcropping Carboniferous granites (Bea, et al., 2004). To the S, the Schist-Greywacke Complex presents NW-SE trending upright folds and faults and a

much moderate volume of granites.

The profile presented in this paper, crosses three geological domains within Central Iberia: the ICS and its Tertiary foreland basins, the DB and TB. The ICS is an intraplate range described as a thick-skin pop-up and pop-down configuration with an E-W to NE-SW orientation, that runs from the Iberian Chain to Portugal (de Vicente et al., 2007, 2018). It started to develop

by the Alpine compression that affects the Iberian Peninsula since the Late Cretaceous. Outcrops in the chain are primarily composed by Variscan granitites with minor outcrops of metamorphic rocks, that correspond to the Variscan basement of the Iberian Peninsula. (Vegas et al. 1990., De Vicente et al., 1996, De Vicente et al., 2007). The profile is enclosed in the western sector of the range, namely, the Gredos sector.

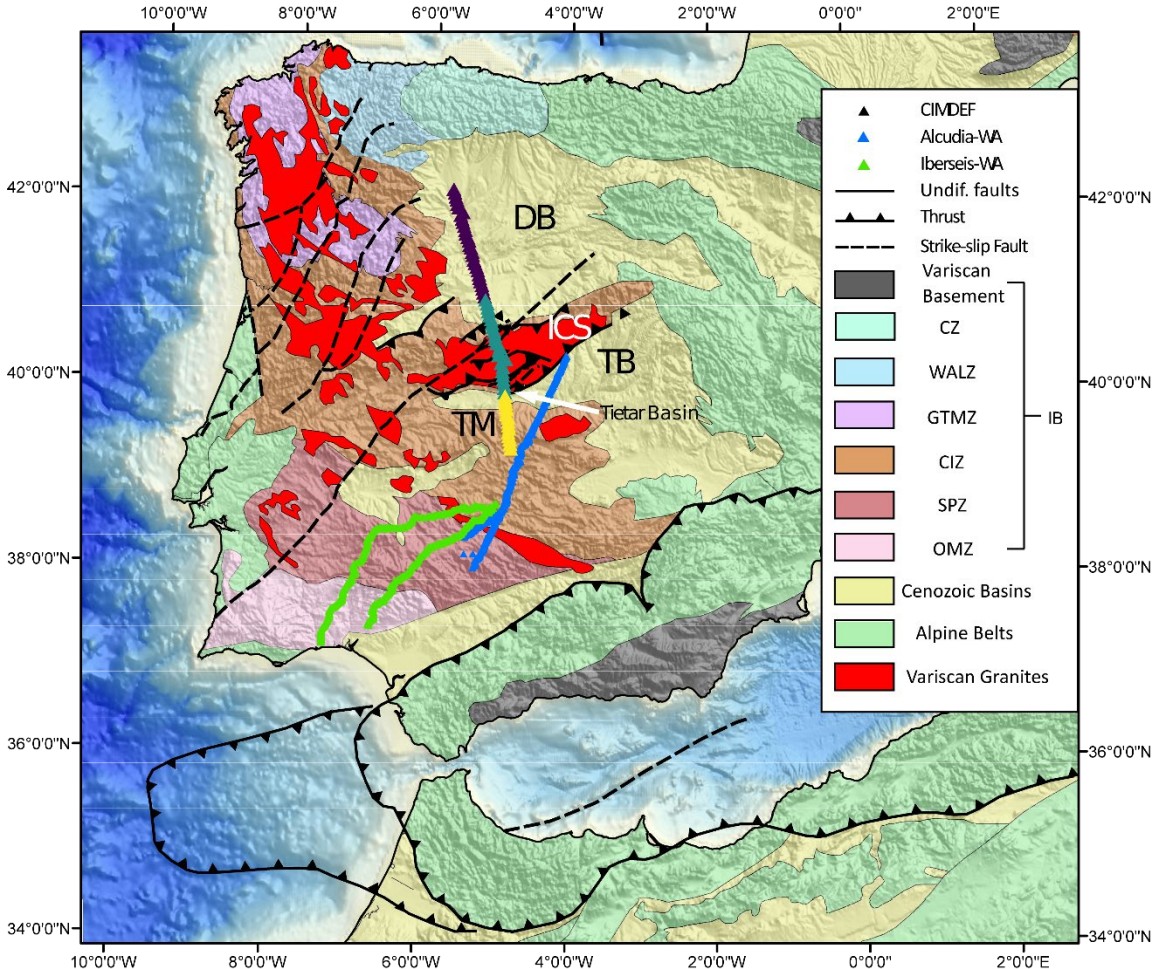


**Figure 1: Simplified geological map of the study area with major tectonic provinces and structures of the Iberian Massif (modified after Andrés et al., 2019)3. Location of the main seismic profiles acquired in the area is also shown. Color coded by deployment, green (1st deployment), yellow (2nd deployment) and purple (3rd deployment). TM:Toledo mountains, ICS: Iberian Central System, DB: Duero Basin, TB: Tajo Basin, CZ: Cantabrian Zone, WALZ: West Asturian Leonese Zone, GTMZ: Galicia–Trás-os-Montes Zone, CIZ: Central Iberian Zone, OMZ: Ossa-Morena Zone, SPZ: South Portuguese Zone. Locations of Variscan granites and granitic zonation are taken from Simancas et al. (2013).**

The knowledge of the lithospheric structure of the ICS and surrounding basins comes from seismic studies (Suriñach and Vegas, 1988, Mancilla and Diaz, 2015) and inversion and forward modelling of potential field data (De Vicente, et al., 2007, Carballo et al., 2015, Torne et al., 2015, Andrés et al., 2018). These results have led to the interpretation of a slightly thickened symmetric crust below the ICS with a Moho depth that deepens from 31 km to 35 km. Lately, Andrés et al. (2019) used autocorrelation of teleseismic data to partly image this change in the Moho depth. These authors suggested that an imbrication



of the crust below the ICS in an asymmetric Alpine-like subduction structure results in a ~7 km Moho offset to the N of the ICS.

## 3 Data and Instrumentation

Data used in this study was acquired within the CIMDEF experiment by 69 short-period (2 Hz), 3-component stations. These were operational during 3 different time periods. The central segment (Fig. 1) of the profile was recorded between May and June 2017 by 24 stations and covered almost 120 km. The second acquisition time was held from February to April 2018 and consisted in a deployment of 15 stations, covering the southern part of the profile. The northern and longest part, almost 170 km, was acquired between July and September 2018 and 30 new stations were installed. The data was acquired in continuous

recording mode at 250 samples per second (sps) during a period ranging from 28 to 60 days depending on the survey. The stations were deployed in a linear array running NW-SE with an interstation spacing of 4,8 km covering a total length of almost 330 km (Fig. 1). All data was collected using the same equipment of sensor and dataloggers, and the same acquisition parameters. While the duration of each deployment was different, the minimum amount of time devoted to recording data was 28 days.

## 4 Method and Data Processing

The methodology used in this study aims to retrieve the Earth's reflection response below single stations by applying autocorrelation of ambient seismic noise. Autocorrelation evaluates the similarity of a seismic trace with a delayed version of itself, whose response depends on the subsurface structure. We have used the vertical component of the data as it is expected to be the one where more P-wave energy is recorded.


The methodology employed for the processing of the continuous recordings include i) pre-processing and ii) construction of stacked autocorrelograms of the vertical component of the ambient noise. We base our processing steps in the Phase Cross-Correlation (PCC) (Schimmel, 1999) and the time-frequency domain phase-weighted stack (tf-PWS) (Schimmel & Gallart, 2007) . The PCC utilizes the instantaneous phases of the analytical signal of the data trace and produces a similarity

measurement of the trace relative to a delayed version of itself. The use of the instantaneous phases makes the correlation amplitude unbiased, which eases the pre-processing as no corrections for high amplitude events have to be applied (Bensen et al., 2007, Schimmel et al., 2011, 2018) The tf-PWS is a linear stack weighted by the time-frequency-dependent instantaneous phase coherency. It enhances the signal by the summation of the envelope normalized analytic signals, strengthening coherent arrivals and attenuating incoherent signals.






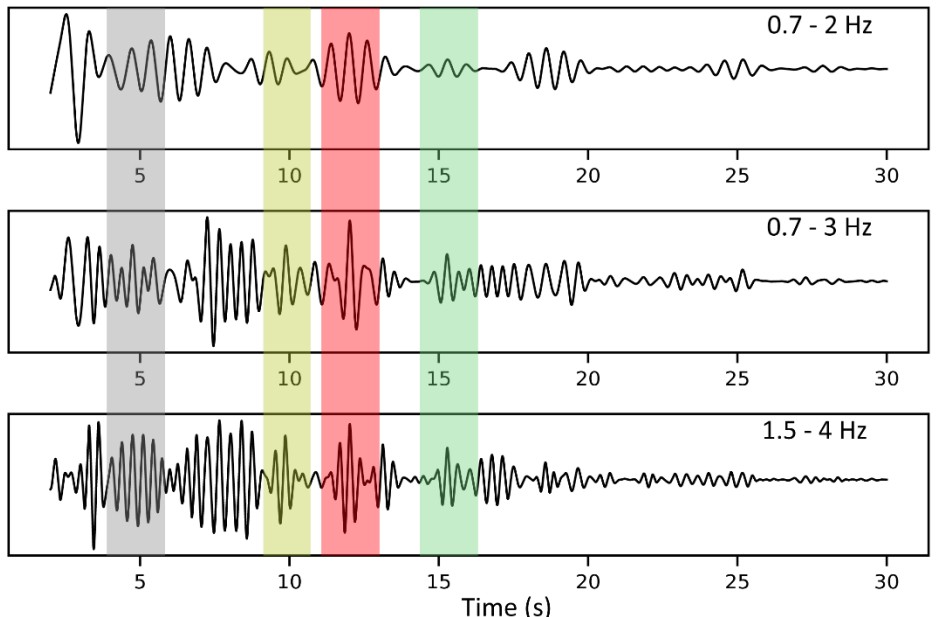

**Figure 2. Example of frequency bands tests, ranging from low frequencies (top) to higher frequencies (bottom). Coherent identified reflections are enclosed in colour bands. It is clear that, as frequency increases, more details are retrieved in the autocorrelations, while consistently capturing the main reflections.**


To assess the quality of the data, it was visually inspected for gaps or anomalous trends. The pre-processing applied consisted in splitting the daily data in 1h long, non-overlapping traces, removing the mean and linear trends and decimating the data from 250 sps to 125 sps. The next step consisted in applying a zero-phase band-pass filter to the data to enhance frequency bands where we expect the target information to be found. There is no general agreement about the best frequency band to be

used in autocorrelation of ambient noise data, although higher frequencies are generally employed to resolve shallow discontinuities which otherwise would be hidden in sidelobes of the zero-lag autocorrelation peak (e.g., Romero and Schimmel, 2018). Different authors have used various sets of frequency ranges for the same purposes. Gorbatov et al. (2013) used 2-4 Hz frequency to retrieve PmP in Australia using autocorrelations of ambient noise. Kennet et al. (2015) utilized 0.5-4 Hz in Australia to image lithosphere-asthenosphere reflectivity. Oren et al. (2017) used lower frequencies (between 0.3-0.55 Hz) to

retrieve body-waves in North America. Recently, Taylor et al. (2016) also utilized low frequencies (0.2-0.4 Hz) in Turkey to retrieve crustal reflectivity. Higher frequencies between 3-12 Hz have been used by Romero and Schimmel (2018) to map the crystalline basement of the Ebro Basin in Spain, but they also showed retrieval of Moho reflections with frequencies of 2-4 Hz. Therefore, it can be argued that the best frequency range depends on the data, structural complexities and the objective of the study. We have tested frequencies ranging from 0.3-0.5 Hz to 1.5-4 Hz to assess the best suited band to retrieve body-

waves for lithospheric imaging in the study area (Fig. 2). The selection of the best frequencies was based on the recovered reflectivity and the consistency of the daily stacks of the stations. The selected frequency band applied to all stations was 1.5-





4 Hz as it provides good reflectivity down to upper mantle depths along with consistent daily stacks for all stations. After applying a band-pass filter, the 1h data segments were autocorrelated for a 0-30s lag-time window using the PCC (Schimmel, 1999). The consistency of the autocorrelograms is checked by plotting the daily autocorrelations together to highlight coherent

160    arrivals (Fig. 3). Where arrivals are consistently retrieved for most of the days, it is considered that they are reflections responding to the subsurface structure. Note that a P-to-P wave reflection at an impedance increase is expected to have a negative amplitude (blue and/or troughs) owing to the free-surface reflection. The data is then daily stacked using the PWS and finally the daily stacks are summed to get the final autocorrelogram. Through stacking autocorrelograms we obtain the P-wave reflectivity of the subsurface. The autocorrelation process creates at t = 0 s a strong arrival which ideally corresponds to

165    a delta function (Claerbout, 1968) whose sidelobes dominate and obscure the early time of the autocorrelation trace. The sidelobes are due to the convolution of the zero-lag delta pulse with the effective noise source time function. To eliminate this effect and for visualization purposes, we have muted the signal from 0 to 3 s, and data has been amplitude normalized.

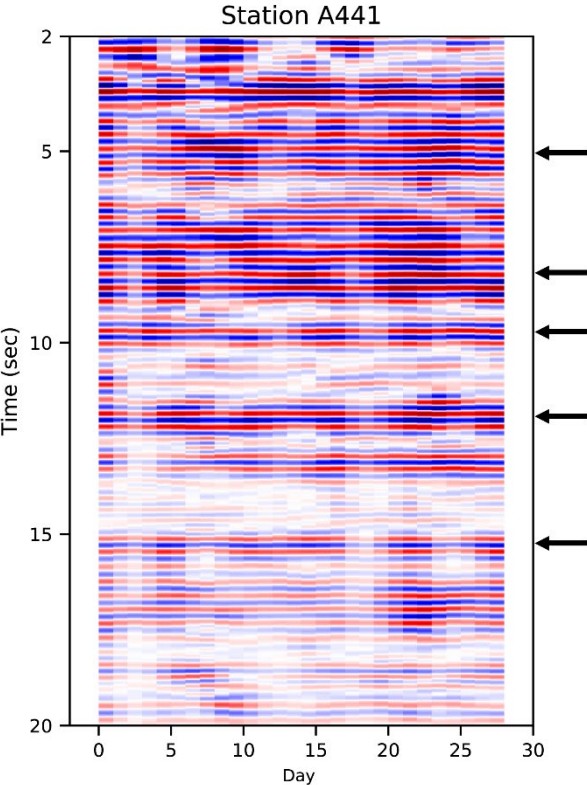

**Figure 3. Daily autocorrelation section for station A441 with arrows showing examples of consistent events. Autocorrelations were**
170    **computed using phase cross correlation within a frequency band between 1.5- to 4-Hz. Blue and red colours mark the positive(peak)**
**and negative(trough) amplitudes, respectively.**



# 5 Results

The procedure described above, led us to obtain a P-wave reflectivity profile (Fig. 4a), crossing the ICS, the DB and the TB. It can be regarded as the reflectivity of the upper lithosphere down to 30s two-way travel-time (TWT). To estimate the approximate depth at which reflections are present, we adopted the same time-to-depth conversion used by Andrés et al. (2019). This conversion takes as a reference the velocity profile of shot 3 from the ALCUDIA wide-angle (WA) experiment Fig. 1, Ehsan et al., 2015) for crustal velocity while for sub-Moho reflections a constant velocity of 8 km/s has been used. In areas with a sedimentary cover or where the crust is thicker than that of the ALCUDIA-WA shot 3 section, the conversion might not be accurate because of the lower velocity in sediments and the crust, thus resulting in an overestimation of the depths at which reflectors are present. The profile covers a distance of approximately 330 km and the interstation spacing ($\approx$ 4,8 km) ensures a high lateral resolution of the crustal structure of the study area.

In general, the profile shows bands of high reflectivity and other rather transparent areas (Fig. 4a). It allows us to divide the crust in upper and lower crust, similar to the pattern obtained in the ALCUDIA and previous CIMDEF datasets. The upper crust extends from 0 to 5-6 s TWT, while the crust-mantle boundary is located between 10-12.5 s TWT. The deepest reflection present in the profile is located between 18-19 s and it is visibly scattered throughout the array. The profile can be divided into three areas on the basis of its geological interpretation: northern, central and southern segments. The northern sector comprises 30 stations, covering almost 140 km, and is crossing the Tertiary DB. The central segment covers the core of the ICS (around 120 km) and presents the highest structural complexity of the profile. The southern segment is the shortest and covers only 72 km, crossing part of the TB/Tietar River Basin (TRB) and the CIZ.

In principle, we cannot rule out the presence of S-wave reflections or P-to-S reflection conversions. These waves are not expected to have a high amplitude on the vertical components, but can occur due to lateral heterogeneities and to an ambient noise wave field which is not diffuse. Here, we interpret dominant signals as P-waves and assume that S-waves are much weaker on the vertical components, and thus, not considered here.





**Figure 4. A) Reflectivity profile retrieved by autocorrelation of ambient seismic noise. In the wiggle plots, the grey lobes indicate**
**negative polarity. Coloured triangles placed over a topographic section represent the different acquisition stages (from N to S: third,**
**first, and second deployments). B) Interpretation of different reflectors, which are labelled between A-F. A marks the boundary**
**between the upper crust and the lower crust. B represents the interpreted depth extension of granites below the ICS. C marks a**
**intracrustal reflector within the lower crust. D is the crust–mantle boundary. E represents a key S verging thrust affecting southern**
**end of the ICS. F marks the scattered reflectivity within the upper mantle.**

In general, the profile shows bands of high reflectivity and other rather transparent areas (Fig. 4a). It allows us to divide the

crust in upper and lower crust, similar to the pattern obtained in the ALCUDIA and previous CIMDEF datasets. The upper

crust extends from 0 to 5-6 s TWT, while the crust-mantle boundary is located between 10-12.5 s TWT. The deepest reflection

present in the profile is located between 18-19 s and it is visibly scattered throughout the array. The profile can be divided into

three areas on the basis of its geological interpretation: northern, central and southern segments. The northern sector comprises





30 stations, covering almost 140 km, and is crossing the Tertiary DB. The central segment covers the core of the ICS (around 120 km) and presents the highest structural complexity of the profile. The southern segment is the shortest and covers only 72 km, crossing part of the TB/Tietar River Basin (TRB) and the CIZ.

In principle, we cannot rule out the presence of S-wave reflections or P-to-S reflection conversions. These waves are not expected to have a high amplitude on the vertical components, but can occur due to lateral heterogeneities and to an ambient noise wave field which is not diffuse. Here, we interpret dominant signals as P-waves and assume that S-waves are much weaker on the vertical components, and thus, not considered here.

## 5.1. Northern segment

The northern sector (Fig. 4a) covers a distance of 140 km and lies entirely within the Cenozoic DB. In general, good reflectivity is recovered down to almost 20 s TWT, despite the fact that the first 8 stations to the N present poor reflectivity below 5-6 s TWT. Nonetheless, several bands of reflectivity within the crust are observed. First, an upper band of reflectivity appears at 5-6 s as a high amplitude event. Below this reflection, another less reflective band appears to be limited by a strong reflector at around 8 s. Further down, at ~9.5-10.5 s, a sharp continuous reflection can be identified along the entire section. Below,
reflectivity and coherency decrease, and only two more sets of reflections seem to be visible throughout the segment. These events are enclosed between 12.5 and 14 s TWT. Finally, some local reflections appear, e.g. one at 18 s TWT, but they are only detected below certain stations.

## 5.2. Central segment

The central segment hosts the most complex reflectivity pattern. Nevertheless, clear events are identified at various depths
(Fig. 4a). First, a package of clear reflectivity is found between 3 to 5-8 s, slightly thinning towards the N and S limits of the ICS. It is characterized by higher amplitudes and lower frequencies compared to the reflectivity seen elsewhere, where higher frequencies are found. The boundary of this reflective area is deeper in the centre of the ICS (8 s TWT) and shallower towards N and S, getting up to ~3 s in the upper part of the section. It also presents higher but less coherent amplitude towards the northern sector while decreasing towards the S of the ICS. Between 200 and 250 km along the profile, the bottom of this band
of reflectivity is poorly defined as more events are probably interfering. Nonetheless, its lateral continuity is easily identified.

An interesting feature located underneath this upper band of reflectivity is the contrasting signature and opposite polarity of events to the N and S of the 200 km point. Between 160 and 200 km, the sub-horizontal low amplitude reflectivity located at ~10 s contrasts with that found between 200 and 250 km, where a slightly N dipping, high-amplitude and high-frequency
package of reflectivity appears. The high amplitude reflectivity observed at ~10. s to the N of the ICS, appears to sink down to ~12.5 s in the central part of the segment. This band of reflections, which is subhorizontal elsewhere, is characterized by a marked lateral continuity throughout the profile. However, underneath the northern part of the ICS, this event presents a N





dipping attitude. Another package of high-amplitude reflections parallel to the one above described but 1-1.5 shallower (E in Fig. 4) is found along the section, also in the southern and central parts of the ICS, at 11 to 11,5 s.


At later times, two sub-parallel reflectors are found at 12.5 and 13.5 s TWT, and are visible at both ends of the segment but not in the central part. Below these reflections, the coherence of the reflectivity decreases. Still, some tentative events can be followed locally at 15 s TWT and between 18-19 s TWT.

### 5.3.    Southern segment

The southern segment represents the shortest section of the profile, approximately 70 km locataed S of the ICS. The deployment was undertaken during an anomalous period of heavy rainfall in the area. Variations of seismic velocities have been observed due to the presence of highly saturated media in shallow layers (Sens-Schönfelder and Wegler, 2006; Obermann, et al., 2014, Fores, et al., 2018). According to previous studies, high rainfalls can be related with de-coherence in the retrieved signal, and increasingly at higher frequencies. During the deployment period, precipitations two to three times above the average took

place. As a consequence, the coherency of the calculated autocorrelations is worse compared to the other two deployments, i.e., reflections are weaker and more noise seem to be retrieved. Altogether, this makes the assessment of the retrieved autocorrelations difficult to interpret and results should be taken with caution.

In this context, the main reflections found in the Central Segment (5-6 s and ≥10 s reflections, Fig. 4a) can be followed to the

S. The 5-6 s reflection has a rather flat geometry shallowing towards the S, getting up to 5-4,5 s. The reflection at around 10 s is more difficult to recognize although the one parallel to it but shallower (8.5-9 s TWT) is a bit more conspicuous. In any case, clear reflections of both events are retrieved just at a few stations, but not continuously along the array. Moreover, their amplitude and coherence is low. Finally, reflectivity underneath 10.5 s is scarce and difficult to correlate.

### 6 Discussion

The interpretation of our data relies on the identification of arrivals with high amplitudes, lateral coherence, and similar waveforms as those on their neighbouring stations. In addition, it is supported by previous knowledge of geological/geophysical features of the study area. In this regard, the ALCUDIA Wide-Angle (WA) profile (Ehsan, et al., 2015) located to the S (Fig. 1) is especially helpful as its resulting velocity model can be used to carry out the depth conversion of main features. Also, we have used constrains of the recently published CIMDEF coincident transect, where interferometry of

earthquake phases with epicentral distance >120° was applied (Andrés et al., 2019).

In general, clear reflectivity and good lateral continuity is retrieved along the profile presented here (Fig. 4b). Unfortunately, to the SE, where higher frequency data could indicate the existence of more heterogeneities and small-scale features, noise due





to weather conditions hinders a detailed interpretation. Nevertheless, the resulting profile allow us to identify crustal and even
upper mantle reflectivity. The crust could be divided into upper and lower crust according to the present reflections. The upper
crust extends from the surface to ~5-6 s TWT where a clear crustal scale discontinuity is observed (Fig. 4b, line A, Fig. 5).
The lower crust extends down to 12.5 s TWT in its deeper part with the Moho being marked by a discontinuity placed between
9.5 and 12.5 s TWT (Fig. 4b, line D). Below this discontinuity, within the upper mantle, lower amplitude scattered events are
present at different times, *e.g.,* 12-13 s TWT and 18-19 s TWT (Fig 4b, line F). As expected, the upper mantle presents less
reflectivity than the crust.

## 6.1. Crustal features

As described above, two crustal scale discontinuities are identifiable in the reflectivity profile (Fig. 4b and 5): a mid-crustal
discontinuity and the crustal-mantle boundary. Moreover, a conspicuous package of high reflectivity is observed in the lower
crust, between 7-8 s TWT (Fig. 4b, line C), in some parts of the profile.

### 6.1.1.  Upper crust

The upper set of reflections is related to the upper-lower crust boundary. It appears mainly at 5 s TWT, being shallower towards
both ends of the profile and slightly deeper below the ICS, mostly in its southern part (~6 s TWT). It constitutes a high
amplitude reflection which underlies an upper crust featuring heterogeneous reflectivity. This boundary is well correlated with
a mid-crustal discontinuity interpreted as the upper-lower crust boundary by Andrés et al. (2019) along this same transect and
with a similar one identified in the ALCUDIA dataset (Ehsan et al. 2014; 2015; Martínez Poyatos et al. 2012). In all cases,
this upper-lower crust boundary coincides in geometry, depth and regional extent character. A little discrepancy exists below
the NW end of the ICS, where the boundary, as identified by GloPSI, is slightly deeper than that presented in this work
(compare Fig. 5 in this paper and Fig.5 in Andres et al, 2019).
The upper crust presents a heterogeneous response throughout the profile, being the central segment where the highest
amplitude events are localized. The outcropping rocks of the ICS are mainly Carboniferous granites (Bea, 2004). In fact,
previous studies have inferred that granites below the ICS exist down to ~18 km based on the low frequency/high amplitude
reflectivity retrieved above that depth (Andrés et al., 2019). Nonetheless, in our profile we found two distinctive zones within
the upper crust below the ICS (Fig. 5), i) from 210-240 km, i.e., the central part, ii) the prolongation to the N and S boundary
of the ICS. The latter is defined by the edge of the low frequency/high amplitude reflections (Fig. 4b, lines B and E). When
considered as a whole, the ICS area defined here would correlate well with that identified with GloPSI as the upper granitic
crust (Andres et al. 2019). However, the higher frequencies used in this study allow us to improve this interpretation. In the
central part of the granitic area (210-240 km), a pattern of higher frequency/higher amplitude reflections is found, which
extends down to 8 s TWT (Fig. 5), cross-cutting the proposed upper-lower crust boundary. However, towards both edges, this





pattern of reflectivity occupies shallower zones, showing N and S dipping contacts with the neighbouring areas. Given these two zonations, we can infer that, the granitoids representing the ICS reach down to 8 s TWT (Fig. 4b, lines B and E) in its deepest point, shallowing to 3.5/4 s TWT towards the S and N. To the N, this characteristic granitic signature fades when entering the DB, as according to our section, it partly overlaps the granites. To the S, this package of reflectivity is bounded

by an area of lower reflectivity that could define some sort of ICS southern thrust. In addition, it is also overlapped by the TB. Finally, a loss of coherency in the autocorrelations marks the end of the granites although the existence of granitic rocks outcropping further to the SE of the CIMDEF profile (E of the Toledo Mountains, TM in Fig. 1) might indicate that they somehow continue in depth, irregularly distributed underneath the TB. Unfortunately, the lack of information above 3 s TWT hinders a more accurate interpretation of the geometry of these granitoids.


The ICS granites represent a large volume of Carboniferous melts derived from crustal thickening and extension during the Variscan Orogeny (Pérez Estaún et al., 1991). The fact that the base of the granites presented in this work extends below the proposed mid-crustal discontinuity (Fig. 5) could indicate that melting of the crust below the central part of the ICS was greater than in other areas of the CIZ and affected a major portion of the crust, partly including the lower crust. In fact, some authors

have shown the major presence of I-type granites in the ICS (Villaseca et al. 2017) suggesting that they have deep photoliths.

It is not clear how the faults controlling the pop-up/pop-down structure (de Vicente et al., 2007) of the ICS affect its deep configuration. They probably play an important role in defining the northern and mostly the southern limit of the ICS (Fig. 4b lines B and E). In the external areas of the identified ICS granitoids, where the upper-lower crust boundary is deeper than the

granites, the lower amplitude reflectivity below the granites would represent the seismic response of the Schist-Greywake Complex (Vendian to Lower Cambrian metasediments) and underlying rocks. These lithologies extend southwards and dominate the outcropping geology until the southern end of the profile.

As stated above, the interpreted upper-lower crust boundary correlates well with that deduced in Andres et al. (2019) and with

similar reflectors seen in the IBERSEIS and ALCUDIA datasets (e.g., Simancas et al, 2003, 2013; Martínez Poyatos et al., 2013; Ehsan et al., 2014, 2015). Accordingly, this reflection runs from the southern part of the SPZ, along the OMZ and up to the southern part of the ICS in the CIZ, i.e., it has a length of around 500 km. This boundary has been proposed to represent the brittle/ductile transition (Simancas, et al., 2003, Martínez Poyatos, et al., 2012, Ehsan, et al., 2014), being the boundary between two crustal levels where tectonic shortening is resolved by different mechanisms (Simancas et al., 2013). In the

CIMDEF profile, the heterogeneous seismic signature of the upper crust seems to picture some dipping reflectors while the lower crust presents mostly sub-horizontal and laterally coherent features. This different response, partly observed by Andrés et al. (2019) might be the representation of two decoupled zones, making the correlation between this reflection and the brittle/ductile transition also feasible. However, due to the fact that ductile deformation exists above this boundary, we suggest that this reflector mostly represents a detachment, probable the rheological boundary between a Pre-Variscan basement and





the deformed rocks on top. According to the interpretation included in this paper, the development and emplacement of the ICS granites erased the imprint of this detachment underneath this mountain range. Accordingly, it has not been active in this area since the Late Carboniferous, thus imposing some constraints to the Alpine accommodation of the deformation that will be discussed later.

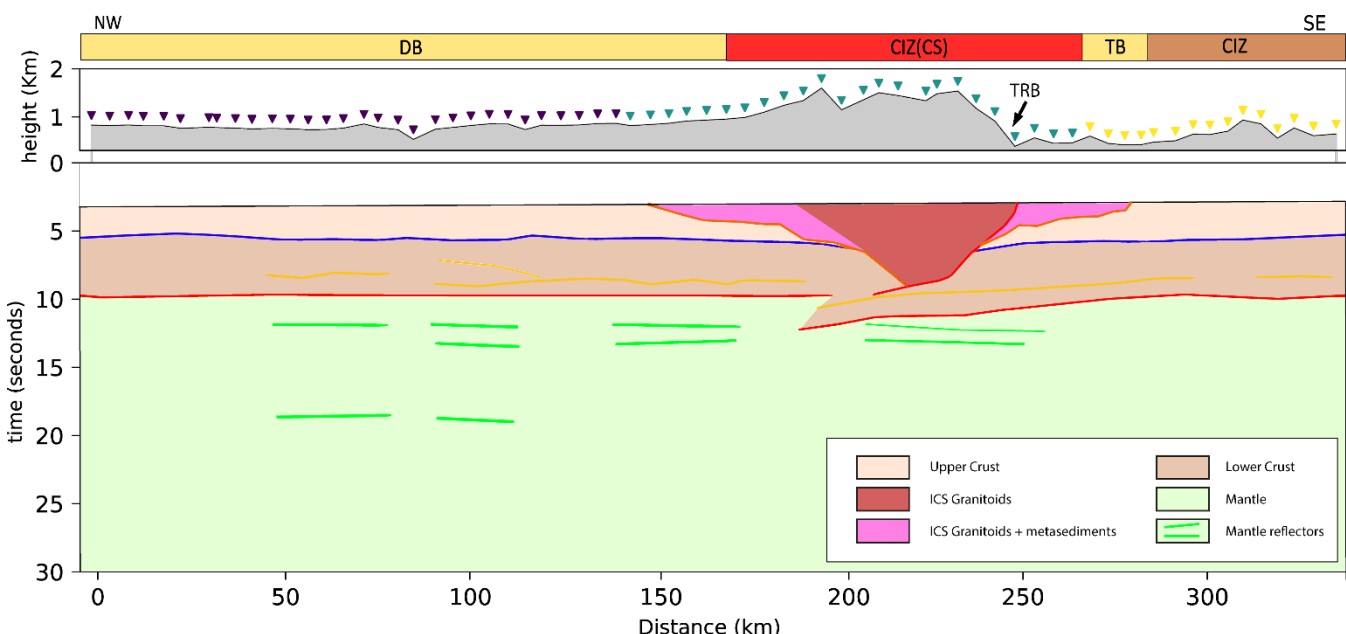

**Figure 5. Sketch of the proposed crustal structure below the Iberian Central System, and the Duero and Tajo Basins. Light and dark brown and represents the upper and lower crust respectively. Light red represents the extend of the granites below the ICS while dark red corresponds to the melted crust. Pale green indicates the upper mantle. Coloured lines are the same as in Fig. 4b.**

### 6.1.2. Lower crust

The lower crust is characterized by some thick sub-horizontal and coherent packages of high-frequency/high-amplitude reflections that seem pretty continuous and well defined at places. The northern sector is less reflective, being almost transparent between 0 and 40 km along the profile. Further to the S, high-amplitude sub-horizontal events are retrieved between 7-8 s TWT, e.g. from 40 to 75 km, from 90 to 160 km and from 190 to 280 km along the profile. Those features could represent a highly laminated lower crust, similar to that observed southward of this profile (e.g. ALCUDIA-NI and IBERSEIS-NI).

Again, this same reflectivity is also imaged by Andrés et al, (2019), after autocorrelations of teleseismic data. The fact that the lower crust is less reflective in the northernmost part of the profile underneath the DB indicates either a change in the nature of the lower crust (i.e. the pre-Variscan basement), a different tectonic evolution for this area or a stronger absorption of P-wave energy by the DB sediments. In fact, reflectivity is low at every level below 5 s TWT in the first 40 km of the seismic profile thus suggesting that attenuation of waves energy may play a key role. On the other side, some cross-cutting relationships

can be addressed at lower crustal levels in this section (e.g., ~100 km). Even though this dataset does not have enough



resolution, other authors have seen the same features in vertical incidence datasets further S (ALCUDIA-NI and IBERSEIS-NI) and have interpreted it as the tectonic Variscan imprint on the lower crust pre-Variscan deformation (Ayarza et al., this volume).

Probably, the most relevant reflection in this profile is the crust-mantle boundary. Previous studies based on gravity modelling (De Vicente et al., 2007), and receiver functions (Mancilla and Diaz, 2015) have proposed that the Moho underneath the ICS has a synform-like structure, with a gentle thickening of 2-3 km. However, Andrés et al. (2019) proposed that there is an imbrication of the lower crust below the ICS, where the southern CIZ lower crust underthrusts that to the N, defining a Moho offset underneath the highest peaks of the mountain range. In our study, the Moho is presented as a distinctive and sharp
reflection identified between ~9.5 s and 12.5 s TWT (29-38 km depth). The boundary between the crust and the mantle presents a rather flat geometry, being shallower to the N, around 9.5-10 s and deepening below the ICS, although shallowing again towards the S. In the central part, a set of north-dipping features can be inferred between 10.5 and 12.5 s. These reflections run from the southern end of the central segment until the centre of the ICS and represent the base of the reflective crust. Therefore, this new image pictures a Moho boundary featuring an offset compatible with an imbrication of lower crust underneath the
ICS. This structure defines the ICS crustal thickening, picturing an asymmetric crustal structure for this mountain range. The crustal thickening identified here appears at the same location as in Andres et al. (2019) thus supporting their interpretation. The underthrusted lower crust, would account for the crustal root seen in gravimetric models although with a different geometry, which has some geodynamic implications. The time offset observed in the Moho is ~2-2.5, s which translates to a thickening of around 6-8 km. The shortening addressed by this crustal imbrication seems ~20 km, although it could be larger.

Comparing the profile and interpretations presented in this study (Figs. 4 and 5) with those generated with earthquake recordings (Andres et al. 2019), the same Moho offset can be identified. However, some differences are observed as the resolving power of both datasets is different. The present data shows that, reflections picturing the imbrication partly define a staircase configuration, describing a smooth underthrusting. Crustal thickening is defined by the signal retrieved from five
stations, from 185 to 210 km approx., and not only by two or three, like in Andres et al. (2019). Moreover, now we can observe another reflection at ~10 s TWT in the root zone. This feature is clearly visible below the three stations that recover the deepest part of the imbrication, from 185 to 200 km. These stations present two reflectors, one located at ~10 s TWT and another located at ~12.5 s TWT, clearly defining the Moho offset that affects the internal parts of the lower crust. We suggest that this pattern is identifiable due to the higher resolution of the data used for the study. The frequency band applied to the data, might
not resolve well the complexity of the structure at hand and its spatial extension but it does image an imbrication whose lateral extent (around 20 km at lower crustal level) could be used to compare with that estimated from field data at the upper crust. High conductivity at lower crustal level found by Pous et al. (2012) at this same location, is consistent with the existence of such an imbrication affecting the entire lower crust, as interpreted here.



The imbrication of the lower crust below the ICS is an expected consequence of the compression that occurred during the Alpine Orogeny within the Iberian Peninsula, driven by the collision between Eurasian and African plates. This compressional stage reactivated normal faults formed during the late orogenic Variscan extension, and made them play a crucial role in the formation of the ICS. These faults created the pop-up/pop-down structure that configures the topography of the ICS (de Vicente et al., 2007). In its south-western border, the Tietar River Fault and the Southern Central System Thrust may play a crucial role in the crustal structure of the area and the topographic differences between the DB and the TB, as they may be involved in the crustal imbrication process (Andrés et al., 2019). In our image, the upper part of the underthrusting lower crust is well defined and seems to continue upwards, to at least, the lower part of the granitic complex defining the mountain rage (km 220). From there, it continues, still at depth, along the southern edge of the ICS, linking upwards with what we interpret as the depth continuation of the Tietar River fault (E in Fig. 4). As the ICS granitoids have assimilated the mid crustal detachment in this area, we argue that some of the faults affecting the upper crust (e.g., the Tietar River Fault), do not root in this detachment but continue to depth, thus forcing the upper crust to sink. This would imply that the upper crust to the S of the ICS is very slightly underthrusting it, lowering the topography and thus defining a contrast in altitude between the two ICS foreland basins, located to the N (DB) and S (TB) respectively, with the TB being ~300-400 m lower. Therefore, the depth extent of the granitoids implies the local lack of the mid-crustal detachment that would prevent the CIZ upper crust located to the S of the ICS from underthrusting, as suggested by Andres et al. (2019). Contrarily, if the detachment has been assimilated, upper crustal fractures can find their way into the lower crust thus allowing the upper crust to sink. This model is partly in conflict with the seismic images obtained for transects along the TB further to the E (de Vicente et al. 2013), which do not show evidences of upper crustal underthrusting. However, they show some downward bending of sediments in the contact between the TB and the ICS. In addition, the present dataset does not cross the NE-SW oriented TB but the E-W Tietar River Fault (Fig. 1) and the configuration might be slightly different. An estimation of the shortening at upper and lower crustal levels implied by each of the involved structures could help us to support this interpretation.

Previous estimates of the shortening accommodated by the ICS in Iberia based on field observations portrait a minimum of 5-9 km and a maximum of 20 km (de Vicente & Muñoz-Martín, 2012, de Vicente et al., 2018) These values suggest that the amount of shortening observed at upper crustal level is similar to that imaged at the lower crust in Figs. 4 and 5, thus supporting our model. However, other possibilities exist and cannot be ruled out.

### 6.2. Upper mantle

Within the upper-mantle, locally continuous and well defined reflections are retrieved (Fig. 4b, line F and 5). They are observed within the northern and central sector, while in the southern sector less coherent mantle reflectivity is retrieved. These reflections are found within the profile at two main levels: between 13 and 14 s TWT (40–45 km) and between 19 and 20 s TWT (~70 km). The upper set is observed at the northern and central segments, while the deepest reflections are only found in restricted areas between 50 and 120 km and 200 to 250 km along the profile. The top package of reflectivity is composed



by two subparallel and sub horizontal reflections separated by ~1 s TWT and slightly deepening towards the S. The second set of reflections is also tilted towards the S but it is less continuous.


In southern Iberia reflectors at depths similar to our deepest feature have been reported previously (Ayarza et al., 2010, Martínez Poyatos et al., 2012, Andrés et al., 2019), e. g. between 61–72 km in the IBERSEIS profile. These reflections have been proposed to be related to a mineral phase transition from spinel-lherzolite to garnet-lherzolites within the upper-mantle: The Hales discontinuity (Hales, 1969). The coincidence in depths between previous observations and the deepest of the

reflections found in this work suggest a correlation between them, implying that the Hales discontinuity might also exist below the ICS and surrounding basins. No interpretations exist so far for the upper band of mantle reflectivity. Travel time estimations of P to S conversions and multiples indicate that they are not related to those processes. Further modelling is necessary to interpret these reflections. However, that is, as yet, above the scope of this paper.

**Conclusions**

In this study, we present a new lithospheric model of the Central Iberian Zone within the Iberian Massif constructed from autocorrelations of ambient seismic noise, as part of the CIMDEF project. The present work shows that this methodology has potential to provide key constraints in orogen scale studies and complements previous results obtained along the same transect by autocorrelation of teleseismic data. Resulting models have important implications on the understanding of the accommodation of intraplate deformation during the Alpine reactivation of Central Iberia. The profile runs through three major

geological features within central Iberia, namely the Iberian Central System, and its foreland Duero and Tajo basins. Our results highlight a crust divided between an upper part, that is in average 15 km thick, and the lower crust with thickness between 15 to 18 km. The boundary between both crusts is well defined throughout most of the profile. Within the upper-crust the new dataset has allowed us to map, laterally and in depth, the extension of the granites forming the Iberian Central System. This batholith has an extension along the profile of around 120 km, although only half of it outcrops. Based in this new dataset,

we have estimated a maximum thickness of 8 s TWT, equivalent to 24 km for these granitic batholit, although it gets thinner towards its N and S boundaries.

The most important findings of the study are i) the presence of an imbrication of the crust below the Central System and ii) the assimilation of the orogenic scale mid-crustal detachment by the granitoids of the Iberian Central System. The former defines a crustal root, where the Central Iberian Zone crust to the S of the Central System underthrusts this mountain range depicting a Moho offset of ~6-8 km, from ~30 km to 36-38 km. The disruption of the mid-crustal detachment by Carboniferous

crustal melting at the Central System allows upper crustal fractures to reach deep levels into the crust, as they are not forced to root in the detachment. This implies that the imbrication of the crust is not restricted to its lower part but also affects the upper crust. Accordingly, the crustal thickening appears to be bounded to the S by a system of interconnected fractures that might outcrop at the Tietar River Basin, or further to the E at the Southern Central System thrust. In fact, a conspicuous area

of high-reflectivity/high-amplitude events interpreted as granitoids (with variable but probably low amounts of metasediments) are found to bound with the main faults, Although it is not yet clear how much of the upper crust is affected by this deformation, this configuration would force it to sink, thus explaining the lower topography that the Tajo Basin, to the S of the Central System, has when compared with that of the Duero Basin, to the N of this mountain range. Estimated shortening at upper and lower crustal level suggest that this model is coherent and does not need an important amount of upper crustal underthrusting

to explain the ~300 m difference in topography between both foreland basins. Within the upper mantle, scattered reflectivity is found below the northern and central segments of the profile. This define two bands enclosed in depths of 40-45 km for the top band and around 70 km for the bottom one. The geometry of these reflectors is relatively flat, however, a slight deepening towards the S is visible below the Iberian Central System. We describe the deepest of these features as part of the transition zone from spinel-lherzolite to garnet-lherzolite, known as the Hales discontinuity, a regional scale discontinuity already

described towards the S of the profile in other studies.

*Data availability*. Data are available in the Labsis repository by selecting the corresponding year for each deployment. For access to the data, contact Juvenal Andrés or Ramon Carbonell.

*Author contributions*. JA, MR, IM, and PA acquired the data. JA processed the data. JA prepared the article. All authors have contributed to the discussion and article review.

*Competing interests*. The authors declare that they have no conflict of interest.

*Acknowledgements*. The data used for the research carried out in this contribution are stored at the DIGITAL.CSIC data repository. We would like to acknowledge the ICTJA-CSIC Seismic Laboratory (http://labsis.ictja.csic.es/, last access: 11 November 2019) for making their seismic station available for this experiment.

*Financial support*. This research has been supported by the Spanish National Research Program (grant nos. CGL2014-56548-
P and CGL2016-81964-REDE), the regional government of Castilla and León (project SA065P17), and the Generalitat de Catalunya (grant no. 2017-SGR-1022) and the Salamanca University Program for Research Groups; Juvenal Andrés is supported by FPI (Formación de Personal Investigador) from the Spanish Ministry of Science, Innovation and Universities (grant no. BES-2015-071683).






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
