# Peer review of "What can seismic noise tell us about the Alpine reactivation of the Iberian Massif? An example in the Iberian Central System"

_Solid Earth, 2020_

## Referee Comment (RC1) · Anonymous Referee #1 · 22 Jul 2020

This manuscript describes a model for the crustal (nor really lithospheric) structure of the Iberian Central Range on the basis of seismic noise analysis. The main result of the study is the resolution of the structure of the crustal root of this intraplate mountain belt as featuring a thrust fault offsetting the lower crust and Moho discontinuity. The paper is in general fairly written, although I found some points of concern. 1) In the introduction, although there is a lengthy description of the Variscan geology of Iberia, largely irrelevant for the purpose of the ms., little reference is given to the Alpine setting of the Iberian plate interior, in which the study is also framed. 2) The conclusion of the lower crust imbrication was seemingly already reached by Andres et al (2019) in an earlier work, and hence the novel contributions of the ms. appear undermined. 3)

[Figure]

in Fig. 4, the authors should explain how they interpreted the picked reflections, and their uncertainties, e.g. why the crust-mantle boundary is D instead of C (and like that, the attribution of other reflectors). Why granites should be so reflective in the profile? 4) I failed to understand the interpretation of the structure of the upper crust. Clarify the distinction between the ICS granitoids and the ICS granitoids and metasediments, and their boundaries. The caption of Fig. 5 should be rewritten, avoiding qualitative color description and conforming to the actual legend of the figure (e.g. what is melted crust in a present-day section?). The relation of these bodies with "pop-up" structuring is confusing. A "staircase configuration describing smooth underthrusting" sounds contradictory. Where is the mid-crustal detachment shown, and what does it mean "assimilated by granitoids". Can the authors explain better the sentence : if the detachment has been assimilated, upper crustal fractures can find their way into the lower crust thus allowing the upper crust to sink". Detailed comments: l. 90: spell granites l. 250: spell located l. 320: meaning of "photoliths"? l. 420: "Other possibilities exist and cannot be ruled out": Which are these?

---

## Referee Comment (RC2) · Jordi Julià Casas (Referee) · 17 Aug 2020

This manuscript investigates the reflectivity of the Iberian crust along a dense transect crossing the Iberian Central System (ICS) and adjacent foreland basins. Interestingly, the authors utilize autocorrelation functions (ACFs) of ambient seismis noise recorded at a number of short-period (f > 2 Hz) stations. The ACFs are obtained after stacking a number of daily phase autocorrelograms in the 1.5-4 Hz frequency range with phase weight for each station. A reflectivity cross-section is then built from the juxtaposition of single-station ACFs and interpreted in terms of past and present tectonic processes.

Overall, I found the manuscript to be correctly structured, well written and appropriate

for the broad readership of Solid Earth. I have, nonetheless, two main concerns about methodology and interpretation:

(i) Given the large variability in the number of days utilized to construct the final ACFs at different stations (28-60 days), and the relatively small number of days in all of them, shouldn't the stability of the ACFs have been investigated before attempting any interpretations? One way of doing that would be to compute ACFs with an increasingly larger amount of (random) days to see whether the stacked autocorrelograms converge to a stable time series or not.

(ii) The interpretation of the final cross-section (Fig 4b) seems to be strongly guided by a coincident cross-section published by the same lead author, which was obtained from the autocorrelation of telesseismic waveforms. I think it would help to add that cross-section to Fig 4 (as Fig 4c) to better illustrate the choices made by the authors in Fig 4b. In addition, that would also help highlight what 'seismic noise can really tell us about the Alpine reactivation'; without that additional piece of information, it is sometimes unclear whether a specific feature is a new finding from ambient seismic noise or just confirmation of something that was reported elsewhere.

Finally, I have a number of minor concerns/suggestions:

P2L35 - For completeness, I think the following manuscript should be added to the list of lithospheric studies:

Julià, J. & Mejía J. (2004). Thickness and Vp/Vs ratio variation of the Iberian crust, Geophys. J. Int. 156, 59-72.

P6L156-157 - Could the authors be a bit more specific about how the final frequency range (1.5-4 Hz) was selected? Just saying that 'it provides good reflectivity ... with consistent daily stacks' is not very informative.

Figure 3 - Why are the reflections at 2.5 s, 13.5 s and 17 s not selected? To me, they seem as good as the ones with an arrow.

P8L185 - Why is the crust-mantle boundary interpreted at 10-12.5 TWT? I see a stronger reflection at 8-9 TWT along most of the profile; moreover, this feature is not interpreted under the Duero Basin (Fig 4b). Is it for consistency with the telesseismic ACF cross-section?

Figure 4 - I do not see any depth scale in this figure; however, the authors claim the conversion to depth was done (P8L175-177). Could a depth scale be added to Fig 4?

P9L206-P10L218 - This text is duplicated. Please, remove.

P10L237-240 - Could that 'opposite polarity' be some sort of cycle skipping? That region seems to be structuraly complex (Fig 5) and I wonder if a migration to depth would somehow shift the traces enough to place them 'in phase'.

P13L325-326 - Could the location of the 'Schist-Graywake Complex' be indicated in Fig 5?

P15L364 - Perhaps the text after line 365 could be under a new subsection (i.e. 6.1.3. Moho).

P16L420 - What 'other possibilities' exist? As they 'cannot be ruled out', I think those should be explained here.

---

## Editor Comment (EC1) · Irene Bianchi (Editor) · 19 Aug 2020

Dear Authors, The reviewers have raised some major concerns regarding your manuscript on which I agree with and warmly encourage you to carefully address.

In particular, it is not clear how you decide to interpret some data features with respect to others (having an estimate of the error associated to the features helps in this direction, as suggested by Rev#2 ii). This kind of information would give major significance to your work, besides giving solid ground to your interpretation, and it's the missing piece for publication. Moreover it should be clarified which further achievements you obtain about the lithospheric structure thanks to this data processing with respect to

your previous study (Andres et al, 2019 published in this same Journal).

Please also reply and address point by point to all comments of the two reviewers.

---

## Author Comment (AC2) · 10 Sep 2020

We would like to thank Jordi Julià Casas for the revision of the manuscript. The points raised helped improve the quality of the manuscript.

Please also note the supplement to this comment:
https://se.copernicus.org/preprints/se-2020-94/se-2020-94-AC2-supplement.pdf
* * *

---

## Author Comment (AC3) · 10 Sep 2020

We have addressed the concerns raised by the reviewers and have uploaded the a point by point reply to their comments.

---

## Author Response (AR1)

**Reply anonymous reviewer #1**

**This manuscript describes a model for the crustal (nor really lithospheric) structure of the Iberian Central Range on the basis of seismic noise analysis. The main result of the study is the resolution of the structure of the crustal root of this intraplate mountain belt as featuring a thrust fault offsetting the lower crust and Moho discontinuity. The paper is in general fairly written, although I found some points of concern.**

**1) In the introduction, although there is a lengthy description of the Variscan geology of Iberia, largely irrelevant for the purpose of the ms., little reference is given to the Alpine setting of the Iberian plate interior, in which the study is also framed.**

We agree with the reviewer, there is little explanation of the Alpine Orogeny evolution and imprint of the study area. We have modified the geological setting section accordingly. We have changed/added the following in the manuscript:

The geological framework of the ICS is marked by the footprint of two orogenies that affected the area, namely, the Variscan orogeny and the Alpine orogeny. The former, took place during the Late Paleozoic times by the collision between Laurussia and Gondwana (Matte, 2001), which closed the Rheic Ocean and amalgamated these continents along with other minor terranes like Armorica (Franke, 2000; Matte, 2001). These terrains constitute, the  basement of the Iberian Peninsula, and  is composed by Upper Proterozoic to Carboniferous rocks deformed and intruded by granites .

The IB, the Iberian outcrop of the European Variscides,...

The disaggregation of Pangea from the Triassic lead to an extensive period (Ziegler 1990; Van Wees et al. 1998) that formed new plate boundaries. During the Cenozoic, Iberia was enclosed between the African and Eurasian plates. The relative movement of the African plate in NNW direction, compressed the Iberian plate and led to the inversion of previously generate intraplate basins, and the formation of the Pyrenean-Basque-Cantabrian (Dercourt et al. 1986; Cloetingh et al. 2002). When the formation of the Pyrenees was completed, by the Mid Oligocene (Vergés et al. 1995), the deformation was absorbed by the Betic-Rift system and the interior of the Iberian Plate. By Tortonian times, the peak of intraplate deformation was reached (Dewey et al., 1989) and the ICS was generated by the reactivation of previous variscan structures.

**2) The conclusion of the lower crust imbrication was seemingly already reached by Andres et al (2019) in an earlier work, and hence the novel contributions of the ms. appear undermined.**

Certainly, the finding of the lower crust imbrication is not new, but this dataset has allowed us a better characterization of its geometry. In addition, that is just part of the results in the paper, e.g., the constraints on the lateral extension of the ICS granites, the internal structure of the crust, and the identification of a prominent reflector within the lower crust are other important conclusions that were not achieved in the Andres et al. (2019). Aside of not being a new contribution, we believe that further characterizations of the lower crustal imbrication are important as its characteristics have been long debated (from a symmetrical geometry to our models where it appears as an asymmetrical feature, typical of Alpine orogens. Finally, much of the importance of this MS resides in the fact that we have utilized a different energy source as the foundation to build the seismic model, thus supporting the findings in Andrés et al., 2019. The use of this technique to depict crustal scale models is quite new and it is, in itself, an achievement.

**3) in Fig. 4, the authors should explain how they interpreted the picked reflections, and their uncertainties, e.g. why the crust-mantle boundary is D instead of C (and like that, the attribution of other reflectors). Why granites should be so**

**reflective in the profile?**

As stated at the beginning of the Discussion section, we take as a starting point for our interpretation, previous knowledge provided by nearly coincident normal incidence (NI) and wide angle (WA) seismic profiles acquired further to the S also in the CIZ (e.g. Poyatos et al., 2012; Ehsan, et al., 2015, Andrés et al., 2019). These previous results constraint the crustal model in the southern part of our dataset and have proven to be a reliable source of information in order to build up our interpretation, as they rely in well stablished geophysical techniques.

Therefore, some guidelines have been used to interpret the seismic profile. In this case, the Moho arrival was expected at around 10 s TWT, because of the aforementioned previous information, thus making reflection C incompatible with it. The same reasoning goes to the other interpreted reflectors.

According to the high amplitude internal reflections in the lower crust, we agree with th reviewer that it is an enigmatic feature. However, if it were considered as the top of the lower crust, the thickness of this layer would agree with that to the N of the ALCUDIA NI experiment, indicating that this lower crust has been also extended and melted. Our interpretation is that, strong reflections exist at 5-6 s, 8 s and 10 s, the lower crust is partly preserved in its original Variscan thickness (from 5-6 to 10 s

TWT) evidences of local extension and melting exist, thus defining new boundaries at intermediate depths (8 s TWT).

Regarding the reflectivity of the granites, we don't talk about very reflective granites in the manuscript. Instead, we use the changes in the reflectivity signature presents and correlate them with boundaries between different materials. In this case, the granites present a low frequency/high amplitude signature respect with its surrounding materials. The low frequency can be regarded as evidence of the lack of internal structure, which is consistent with the result from active seismic experiments on granitic terrains. However, we have slightly modified this part of the discussion to make it clearer.

"The upper crust presents a heterogeneous response throughout the profile, being the central segment where the highest amplitude events are localized. The outcropping rocks of the ICS are mainly Carboniferous granites (Bea, 2004). In fact, previous studies have inferred that granites below the ICS exist down to ~18 km based on the low frequency/high amplitude reflectivity retrieved above that depth (Andrés et al., 2019). Nonetheless, in our profile we found two distinctive zones within the upper crust below the ICS (Fig. 5), i) from 210-240 km, i.e., the central part, ii) the prolongation to the N and S boundary of the ICS. The latter is defined by the edge of the low frequency/high amplitude reflections (Fig. 4b, lines B and E). This signature can be regarded as the expression of a highly homogenous body and/or with little internal reflectivity. When considered as a whole, ...”

**4) I failed to understand the interpretation of the structure of the upper crust. Clarify the distinction between the ICS granitoids and the ICS granitoids and metasediments, and their boundaries. The caption of Fig. 5 should be rewritten, avoiding qualitative colour description and conforming to the actual legend of the figure (e.g. what is melted crust in a present-day section?). The relation of these bodies with "pop-up" structuring is confusing. A "staircase configuration describing smooth underthrusting" sounds contradictory. Where is the mid-crustal detachment shown, and what does it mean "assimilated by granitoids". Can the authors explain better the sentence: if the detachment has been assimilated, upper crustal fractures can find their way into the lower crust thus allowing the upper crust to sink".**

Regarding the extension of the granites, we have to apology as addressing this comment we have found an error in figure 5. *The marked granitoids + metasediments section to the North, should all be represented as granitoids*. The figure 5 presented in the submitted manuscript is an early version, we apology for this.

[Figure]

*- Clarify the distinction between the ICS granitoids and the ICS granitoids and metasediments, and their boundaries*
The distinction between both zones comes from the reflectivity signature below the ICS. The granitoid + metasediment part presents slightly higher frequencies, and we believe its signature is influenced by a lithology with higher a more complex and reflective internal structure than that of granites. Moreover, the fact that an outcropping fracture (TRB fault) can be continued to that boundary defining a crustal scale feature fracture is present in the limit between the two areas, led us to think that is a relevant boundary there should be some difference. However, this is an interpretation and more information is needed. We have modified the text accordingly.

"To the N, this characteristic granitic signature fades when entering the DB, as according to our section, it partly overlaps the granites. To the S, this package of reflectivity is bounded by an area of lower reflectivity that could define some sort of ICS southern thrust. The reflectivity package south of this thrust present higher frequencies and amplitude. We interpret this change in reflectivity as the seismic expression of a heterogeneous zone, composed by granites and metasediments from the Schist-Greywake Complex, which dominates the outcrops of the southern part of the CIZ. Although this is a feasible interpretation, additional geophysical information is needed to confirm this feature."

*- The caption of Fig. 5 should be rewritten, avoiding qualitative colour description and conforming to the actual legend of the figure (e.g. what is melted crust in a present-day section?)*
We have modified the legend of figure 5 as follows.

"Figure 5. Sketch of the proposed crustal structure below the Iberian Central System, and the Duero and Tajo Basins. Black solid line defines the ICS Southern Thrust. A crustal scale thrust can be defined to the S of the ICS. In its upper part it may coincide with the Tietar River Fault (TRF). It then follows the southern boundary of the ICS and reaches the lower crust, offsetting it to define a crustal imbrication."

We referred as melted crust to the extension of the granites below the ICS. We have eliminated this from the figure caption as it is not melted crust, but the resulting granites from the partial melting of the crust.

*- The relation of these bodies with "pop-up" structuring is confusing*
"It is not clear how the faults controlling the pop-up/pop-down structure (de Vicente et al., 2007) of the ICS affect its deep configuration. They probably play an important role in defining the northern and mostly the southern limit of the ICS (Fig. 4b lines B and E)."
Pop-up/pop-down structures affects the shallower part of the crust as described by de Vicente et al. 2007, but some of the faults that form them can have deep roots, such as the southern thrust as we comment on the paragraph above. As we lack information in the upper 3 s of the profile due to the sidelobes generated by the processing, we cannot discuss further.

*- A "staircase configuration describing smooth underthrusting" sounds contradictory*
We agree and therefore have removed "staircase configuration".

*- Where is the mid-crustal detachment shown, and what does it mean "assimilated by granitoids"?*

We have added to the legend in Figure 5 the mid-crustal detachment and the Moho representation. It runs across the profile but below the ICS its signal has been erased by the granites which reach the lower crust.

    *- Can the authors explain better the sentence: if the detachment has been assimilated, upper crustal fractures can find their*
*way into the lower crust thus allowing the upper crust to sink".*

If the detachment level was present below the ICS that would act as a reological boundary, i.e. a detachment, thus preventing that faults/deformation occurring in the upper crust reached, to the lower crust. The fact that the detachment is not present below the ICS means the contrarily, crustal scale fractures can go deep into the lower crust as we image the ICS Southern Thrust.

**Detailed comments: l. 90: spell granites**

Corrected

**l. 250: spell located**
Corrected.

**l. 320: meaning of "photoliths"?**

We have changed the sentence to:

"In fact, some authors have shown the major presence of I-type granites in the ICS (Villaseca et al. 2017) suggesting that they
have a deep origin."

**l. 420: "Other possibilities exist and cannot be ruled out": Which are these?**

Other possibilities exist as there is not a seismic profile with enough resolution (i.e. seismic reflection data), to assess the amount of shortening that the lower crust has accommodated, although we strongly believe that some of the shortening has
been absorbed by the lower crust below the Iberian Central System. For completeness we have extended this sentence as follows:

"These values suggest that the amount of shortening observed at upper crustal level is similar to that imaged at the lower crust in Figs. 4 and 5, thus supporting our model. However, other possibilities exist and cannot be ruled out. Further deep normal incidence seismic studies with higher lateral resolution are needed in order to assess the amount of shortening accommodated
at lower crust levels."

**Reply reviewer #2**

This manuscript investigates the reflectivity of the Iberian crust along a dense transect crossing the Iberian Central System (ICS) and adjacent foreland basins. Interestingly, the authors utilize autocorrelation functions (ACFs) of ambient seismis noise recorded at a number of short-period (f > 2 Hz) stations. The ACFs are obtained after stacking a number of daily phase autocorrelograms in the 1.5-4 Hz frequency range with phase weight for each station. A reflectivity cross-section is then built from the juxtaposition of single-station ACFs and interpreted in terms of past and present tectonic processes.

Overall, I found the manuscript to be correctly structured, well written and appropriate for the broad readership of Solid Earth. I have, nonetheless, two main concerns about methodology and interpretation:

(i) Given the large variability in the number of days utilized to construct the final ACFs at different stations (28-60 days), and the relatively small number of days in all of them, shouldn't the stability of the ACFs have been investigated before attempting any interpretations? One way of doing that would be to compute ACFs with an increasingly larger amount of (random) days to see whether the stacked autocorrelograms converge to a stable time series or not.

Indeed, it is important to assess the number of autocorrelograms (AC) needed for a stable response in order to be sure that what is retrieved is an actual reflection and not a spurious event or an artefact. We did this by taking daily stack and compare the signal retrieved by each of them. As we sliced every day in pieces of 1 hour we ended up with 24 autocorrelations for each day. Visual inspection in Figure 3 shows that the main reflections retrieved are consistent each day with 24 AC stacked, although minor variability is found, thus assuring that the signal is stable. Stacking more days would yield the same result as we are adding the same signal to the stack. For clarity porpoises, we have added the final stack of the same station used in Figure 3 to compare between the daily stack and the final stack.

[Figure]

**(ii) The interpretation of the final cross-section (Fig 4b) seems to be strongly guided by a coincident cross-section published by the same lead author, which was obtained from the autocorrelation of telesseismic waveforms. I think it would help to add that cross-section to Fig 4 (as Fig 4c) to better illustrate the choices made by the authors in Fig 4b.**

**In addition, that would also help highlight what 'seismic noise can really tell us about the Alpine reactivation'; without that additional piece of information, it is sometimes unclear whether a specific feature is a new finding from ambient seismic noise or just confirmation of something that was reported elsewhere. Finally, I have a number of minor concerns/suggestions:**

We agree that it is a good way to compare previous results with those presented in this manuscript and we have added the GloPSI profile as figure 4c.

[Figure]

Figure 4. A) Reflectivity profile retrieved by autocorrelation of ambient seismic noise. In the wiggle plots, the grey lobes indicate negative polarity. Coloured triangles placed over a topographic section represent the different acquisition stages (from N to S: third, first, and second deployments). B) Interpretation of different reflectors, which are labelled between A-F. A marks the boundary between the upper crust and the lower crust. B represents the interpreted depth extension of granites below the ICS. C marks a intracrustal reflector within the lower crust. D is the crust–mantle boundary. E represents a key S verging thrust affecting southern end of the ICS. F marks the scattered reflectivity within the upper mantle. C) Reflectivity profile extracted from autocorrelations of teleseismic events over the same deployment array (modified from Andrés et al., 2019).

**P2L35 - For completeness, I think the following manuscript should be added to the list of lithospheric studies:**

**Julià, J. & Mejía J. (2004). Thickness and Vp/Vs ratio variation of the Iberian crust, Geophys. J. Int. 156, 59-72.**

Added.

**P6L156-157 - Could the authors be a bit more specific about how the final frequency range (1.5-4 Hz) was selected? Just saying that 'it provides good reflectivity ... with consistent daily stacks' is not very informative.**

The frequency band (FB) was selected after trying different ones, from very low frequencies to higher than 4 Hz (Fig. 2, although only shows until 4 Hz). For each of these FB we tested the consistency of the stacks as explained above. The selection of the best FB was a compromise between resolution and consistency of the produced stacks. As we can see in Figure 2, as we increase the frequency more features are retrieved in the autocorrelation and also appear more defined. As an example, the red box in Figure 2 shows how a single high amplitude reflection, at low frequencies, actually contains part of the signal of another feature that can be resolved increasing the frequency. For instance, we found that above 4 Hz the daily stacks were not converging and we needed more days to begin to find coherence. Accordingly, we decided to select a FB below that top limit and the one preferred was the one explained in the manuscript. Nonetheless we have extended the explanation for clarity.

"Therefore, it can be argued that the best frequency range depends on the data, structural complexities and the objective of the study. We have tested frequencies ranging from 0.3-0.5 Hz to 1.5-4 Hz to assess the best suited band to retrieve body-waves for lithospheric imaging in the study area (Fig. 2). As seen in Figure 2, at higher frequencies the vertical resolution increases. The red box marks a reflection that at lower FB range only resolves a big amplitude event, whereas at higher FB it is clear that it also contains signal from another feature. Therefore, the selection of the best frequencies was based on the recovered reflectivity and the consistency of the daily stacks of the stations. For each frequency band we computed daily stacks of the data and check the convergence of the autocorrelations. For frequencies above 4 Hz the daily stacks did not converge as expected. Due to the limited amount of recorded days, we set the top frequency to 4 Hz. The bottom limit of the filter was set so it avoids the microseismic noise peaks that strongly influences the lower frequencies."

**Figure 3 - Why are the reflections at 2.5 s, 13.5 s and 17 s not selected? To me, they seem as good as the ones with an arrow.**

We agree that those reflections can be marked as some kind of structural reflectivity response. The aim in Figure 3 was to illustrate the consistency of the daily autocorrelations and mark some of the most relevant ones, without bloating the image too much. Moreover, the reflector located at 13.5 s TWT is marked in Figure 5.

**P8L185 - Why is the crust-mantle boundary interpreted at 10-12.5 TWT? I see a stronger reflection at 8-9 TWT along most of the profile; moreover, this feature is not interpreted under the Duero Basin (Fig 4b). Is it for consistency with the telesseismic ACF cross-section?**

As we stated at the beginning of the discussion section, we support our interpretation in previous published results, such as the one mentioned by the reviewer (Andrés et al., 2019), but also by previous wide-angle seismic studies that sample the parts of the same area of interest (Ehsan, et al., 2015) and represent the base upon we build our interpretation. The reflection proposed as the crust-mantle boundary matches with that of Andrés et al, 2019, which provides consistency to the idea of it being the actual Moho interface. Furthermore, the reflection at 8-9 s TWT, would mean a Moho interface at 24-28 km depth (with an average velocity of the crust of 6.2 km/s) which is way shallower than what is proposed in other studies (Ehsan, et al., 2015; Diaz et al., 2016; Palomeras et al., 2017).

**Figure 4 - I do not see any depth scale in this figure; however, the authors claim the conversion to depth was done (P8L175-177). Could a depth scale be added to Fig 4?**

Thanks for pointing out this mistake. The depth scale has been added to Figure 4.

**P9L206-P10L218 - This text is duplicated. Please, remove.**

Removed, thanks for noting it.

**P10L237-240 - Could that 'opposite polarity' be some sort of cycle skipping? That region seems to be structuraly complex (Fig 5) and I wonder if a migration to depth would somehow shift the traces enough to place them 'in phase'.**

Indeed, some problems derived from the structural complexity of the region could arise during the processing. Unfortunately, we don't have a way to assess it as we lack a reliable seismic velocity model to perform a depth migration to the seismic profile.

**P13L325-326 - Could the location of the 'Schist-Graywake Complex' be indicated in Fig 5?**

We have added the extension of the Schist-Graywake Complex as suggested.

Please note that we have added a modified Figure 5 in the revised manuscript as the figure in the submitted manuscript was an early version and not the final one. We spotted this error by a question raised by reviewer #1.

[Figure]

**P15L364 - Perhaps the text after line 365 could be under a new subsection (i.e. 6.1.3. Moho).**

We have adopted this suggestion and named the subsection as "6.1.3. Crust-mantel boundary"

**P16L420 - What 'other possibilities exist? As they 'cannot be ruled out', I think those should be explained here.**

Other possibilities exist as there is not a seismic profile with enough resolution (i.e. seismic reflection data), to assess the
amount of shortening that the lower crust has accommodated, although we strongly believe that some of the shortening has
been absorbed by the lower crust below the Iberian Central System. For completeness we have extended this sentence as
follows:

[revised manuscript text omitted]

---

## Author Response (AR2)

**Reply reviewer #2**

Regarding the use of previous published studies in order to help the interpretation, we have added the following statements in the manuscript. As pointed by the reviewer and the editor it helps clarify why we have interpreted the boundaries as they are and how them correlate with previous studies.

[revised manuscript text omitted]